# Asbestos-Environment Pollution Characteristics and Health-Risk Assessment in Typical Asbestos-Mining Area

**DOI:** 10.3390/toxics11060494

**Published:** 2023-05-31

**Authors:** Xuwei Li, Yun Chen, Xuzhi Li, Mengjie Wang, Wenyi Xie, Da Ding, Lingya Kong, Dengdeng Jiang, Tao Long, Shaopo Deng

**Affiliations:** 1Nanjing Institute of Environmental Sciences, Ministry of Ecology and Environment of China, Nanjing 210042, China; lixuwei@nies.org (X.L.); lixuzhi@nies.org (X.L.);; 2State Environmental Protection Key Laboratory of Soil Environmental Management and Pollution Control, Nanjing 210042, China

**Keywords:** asbestos pollution, risk assessment, risk control values

## Abstract

Asbestos has been confirmed as a major pollutant in asbestos-mining areas that are located in western China. In general, asbestos-fibre dust will is released into the environment due to the effect of intensive industrial activities and improper environmental management, such that the health of residents in and around mining areas is jeopardised. A typical asbestos mining area served as an example in this study to analyse the content and fibre morphology of asbestos in soil and air samples in the mining area. The effects of asbestos pollution in and around the mining areas on human health were also assessed based on the U.S. Superfund Risk Assessment Framework in this study. As indicated by the results, different degrees of asbestos pollutions were present in the soil and air, and they were mainly concentrated in the mining area, the ore-dressing area, and the waste pile. The concentration of asbestos in the soil ranged from 0.3% to 91.92%, and the concentration of asbestos fibres in the air reached 0.008–0.145 f·cc^−1^. The results of the scanning-electron microscope (SEM) energy suggested that the asbestos was primarily strip-shaped, short columnar, and granular, and the asbestos morphology of the soils with higher degrees of pollution exhibited irregular strip-shaped fibre agglomeration. The excess lifetime cancer risk (ELCR) associated with the asbestos fibres in the air of the mining area was at an acceptable level (10^−4^–10^−6^), and 40.6% of the monitoring sites were subjected to unacceptable non-carcinogenic risks (HQ > 1). Moreover, the waste pile was the area with the highest non-carcinogenic risk, followed by the ore dressing area, a residential area, and a bare-land area in descending order. In the three scenarios of adult offices or residences in the mining area, adults’ outdoor activities in the peripheral residence areas, and children’s outdoor activities, the carcinogenic-and non-carcinogenic-risk-control values in the air reached 0.1438, 0.2225 and 0.1540 f·cc^−1^, and 0.0084, 0.0090 and 0.0090 f·cc^−1^, respectively. The results of this study will lay a scientific basis for the environmental management and governance of asbestos polluted sites in China.

## 1. Introduction

Asbestos refers to six naturally occurring fibrous magnesium-silicate minerals, and it has been extensively applied to common industrial fields (e.g., buildings and manufacturing) due to its high tensile strength, prominent chemical degradation resistance, usability, and other characteristics. China’s asbestos mines are primarily located in the northwest and southwest regions, with the Mangya mining region in the northwest generating the most asbestos. There were 1936 asbestos-related businesses in China as of 2019 [1]. Asbestos exposure has been shown to be closely associated with pulmonary fibrosis, pleural plaque, pleural mesothelioma, and lung cancer [2]. Accordingly, asbestos is classified as a Group 1 carcinogen by the International Agency for Research on Cancer (IARC) [3]. The morphological characteristics of asbestos are important factors affecting pulmonary pathogenicity. Asbestos is easily rolled into a cylindrical or “tubular” fibre due to its layered or lamellar silicate structures. Furthermore, its irregular, fine and long fibre can easily cause pulmonary fibrosis. Since asbestos is insoluble in water and migrates heavily through the air with wind, considerable asbestos-fibre dust generated by the disturbance of rocks and soils containing asbestos is released to air in the processes of natural wind erosion [4] and human activities such as mining, processing and use, which may pose health risks to residents in and around mining areas via inhalation [5].

Environmental risk research on asbestos began earlier in Europe and the United States, primarily through qualitative and quantitative assessment methods. These countries attempted to introduce risk-based management methods into the assessment, remediation, and control of environmental risk in asbestos polluted lands [6].The United Kingdom (UK) utilises a qualitative risk assessment for asbestos in soil, which qualitatively evaluates the high, medium, and low risk of asbestos contamination based on plot characteristics and industry categories for subsequent management, with a recommended asbestos threshold value of 0.001%. In the Netherlands, asbestos risk is evaluated using a stratified qualitative risk assessment based on a soil intervention value (100 mg/kg) fitted to simulated experimental and field measurement data; if the investigated value is greater than this value, the risk of spread is considered to be present. Due to the limitations of current testing technology, the United States (US) Environmental Protection Agency (EPA) recommends cleansing measures for soil asbestos levels between 0.25 and 1.00 percent. In general, the health risk assessment of asbestos in a wide range of countries worldwide is mainly based on the use of qualitative methods to establish environmental limits. This lacks the use of quantitative risk assessment for asbestos in environmental media.

Common quantitative risk-assessment models (e.g., the risk based corrective action (RBCA) model in the United States) have insufficient respiratory reference concentration (RFC) parameters, and they are designed to study soil particles instead of fibre dust, such that the risk of asbestos to human health cannot be directly obtained. Over the past few years, the United States has built an Asbestos Risk Assessment Framework (ARAF) for the Superfund, in order to measure the content of asbestos released into the air in the worst conditions. Given the respiration and inhalation-based pathways in a wide variety of exposure scenarios, the risk of asbestos in the air can be quantitatively assessed to provide a novel method for risk assessment for asbestos-polluted sites [7]. As pollution control has been increasingly stressed in China, the content of asbestos (a major pollutant in asbestos mining areas) in the environment and its environmental risks have aroused signifcant attention from environmental management departments. In China, there is still a lack of research on the evaluation of health risks in residential or office settings near asbestos mining areas. Thus, it is particularly important and urgent to carry out environmental monitoring and health risk assessments in asbestos sites.

In this study, the contents of asbestos in various environmental media were determined by collecting soil and air samples from five classes of areas (e.g., a mining area, a ore dressing area, a waste pile, a residence area and a bare land of a typical asbestos mining area located in northwest China). Based on the SEM analysis, a deep insights into the characteristics of asbestos pollution in different area environments was obtained by discriminating the morphology of the fibre in an asbestos polluted source. Using the Asbestos Risk Assessment Framework for the US Superfund Site Survey and localised exposure scenarios and parameters, the human health risk arising from asbestos pollution in the air in residential activity areas in and around the mining area was assessed. This study lays a solid scientific basis for the environmental management and human health risk assessment of asbestos-polluted lands in China.

## 2. Materials and Methods

### 2.1. Overview of Study Area

The study area comprised residence areas within an asbestos mining area and northeast of its periphery. Asbestos-mining area is characterised by a mining history of 62 years, with a length of nearly 6 km from east to west, a width of approximately 4 km from south to north, and an area of about 14.11 km^2^. The mining area was a large ultra-basic rock type chrysotile asbestos deposit, and spiral open-pit mining served as the mining method. In general, asbestos fibres in the soil and air in the mining area were obtained in the natural weathering and artificial mining processes of asbestos mines, as well as tailings and waste residues. Waste piles were distributed around the mining area and south of the mining area, with different stacking areas, heights generally ranging from 3–8 m, and a maximum height of no more than 15 m. The research area featured to a continental climate, which is dry and cold, while exhibiting dramatic changes in temperature and frequent winds in the four seasons. The meteorological data suggested that the annual average temperature of the mining area reached 1.5 °C. The rainfall in the area was sparse, with an annual rainfall of only 46.9 mm. The daily average wind speed generally exceeded 5 m/s, with a maximum of 26 m/s. Notably, no plants were reported in hundreds of kilometers around the mining area.

### 2.2. Sample Collection

In this study, 94 soil samples and sampling points and 57 ambient air-sampling points were assigned to the survey area and then distributed in the mining area, the ore dressing area, the periphery of the waste pile, the residence area, the bare land, and so forth. Since asbestos pollution in the environment largely arises from raw asbestos ores, asbestos tailings, asbestos finished products, and asbestos dust, the four classes of pollution-source samples described above were simultaneously collected in this study for characterization and analysis. Figure 1 presents the distribution of the study area and the arrangement of the sites.

Air samples were collected using an explosion-proof intelligent air sampler GilAir Plus (Sensidyne, United States), with an equipment erecting height of 1.2–1.4 m. The sampler (filter film clamp) equipped with a filter film was connected with a sampling pump, and the entire sampling system was calibrated to ensure that the sampling flow was within a flow range specified in the standard. Next, the sampling time was set based on the calibrated flow to ensure that the sampling volume was not less than the sampling volume required by the detection limit. The sampling flow was set to 1~4 L/min, and the sampling time was at least 8 hours. At the end of sampling, plugs were covered at both ends again, and a rigid bracket equipped with the filter film was transported to the laboratory.

Soil samples were collected with a shovel. Given that asbestos is insoluble in water and easily migrates into air with wind, the sampling depth of soil samples at this time was controlled at 0–30 cm [8] of a surface layer.

### 2.3. Analysis and Quantification Method of Asbestos in Air and Soil

Asbestos in the air was determined using the asbestos fibre concentration detection method of the National Occupational Health Standards of the People’s Republic of China (GBZ/T 192.5—2007) [9], and the filter film was transparently fixed using an acetone vapour method. The fibers were counted with a phase contrast microscope in accordance with the counting rules of asbestos fibres, and the question of whether asbestos fibres observed under the microscope conformed to the counting requirements was more effectively considered by using an ocular micrometer. The counting principle for asbestos fibers is that their length was greater than 5 μm, their width was less than 3 μm, and the length-to-width ratio was greater than 3:1. Lastly, a concentration-calculation equation for asbestos fibres was used to obtain the quantity concentration of asbestos fibres in the air. The equation for counting was as follows: C = A × N/a × n × F × t × 1000(1)
C—Quantitative value of asbestos fiber concentration in the air (f·cc^−1^);A—Dust collection area value of filter membrane (mm^2^);N—The determined total quantity of fibers (f);a—Counting of the field of view of the eyepiece’s micrometer (mm^2^);n—The number of visual fields determined by enumeration;F—Flow sampling value (L/min);t—Sampling time value (min).

Asbestos in the soil and asbestos ores was determined using the Method for the Determination of Asbestos Content in Products (GB/T 23263—2009) [10]. First, the qualitative analysis was conducted using powder X-ray and polarized-light microscope. For samples with asbestos contained, asbestos was quantitatively analysed using the X-ray diffraction analysis method.

### 2.4. Analysis Method of f Morphological Features of Asbestos Fibres

In this study, Hitachi S-4800 field-emission scanning-electron microscopy was used to analyse the morphology of asbestos ore. The main technical indicators were as follows: ① voltage was 2.0 KV; ② working distance was 6.0 mm; ③ signal source was SE2. When measured, material sample were pretreated by crushing, removal of organic matters, preparation of samples, and spraying of conductive ions on their surfaces. Next, scanning-electron microscopy was used to observe the morphology to distinguish fibre particles from other, non-fibre particles.

### 2.5. Assessment Method for Human Health Risk

Based on the U.S. Superfund Risk Assessment Framework, the asbestos health risk assessment in this study considered the exposure pathway of inhaling asbestos fibres via respiration, and the risk of asbestos content in the air was quantitatively assessed [7,11]. After the airborne asbestos content was measured under the conditions of human-interference activities, the effects of airborne asbestos fibre content on human health were assessed in accordance with site characteristics using the excess lifetime cancer risk (ELCR) and hazard index (HQ). The specific calculation is expressed as follows:(1)Carcinogenic risk assessment

Prediction equation for excess lifetime cancer risk (ELCR):ELCR = EPC·TWF·IUR(2)
TWF = daily exposure time/24·annual exposure day/365(3)
where ELCR, EPC, IUR, and TWF denote excess lifetime cancer risk, environmental carcinogenic risk correlated with site-related exposure pathways, exposure concentration, asbestos fibre concentration (f·cc^−1^) in the air under specific activity conditions, inhalation unit risk (f·cc^−1^)^−1^, and time weighting factor, respectively, referring to cumulative discontinuous exposure in a year.

The U.S. EPA suggests that the excess lifetime cancer risk is negligible if it is less than 1 × 10^−6^. Moreover, a risk greater than 1 × 10^−4^ is sufficiently large and requires some repair or control measures. In general, a carcinogenic risk of 1 × 10^−4^–1 × 10^−6^ is acceptable.
(2)Non-carcinogenic-risk assessment

The non-carcinogenic risk assessment is primarily characterised by a hazard index (HQ), defined as a ratio of long-term intake arising from exposure to a reference dose. The EPA suggests that if HQ is less than or equal to 1, it is generally not necessary to repair or apply control measures; if it exceeds 1, non-carcinogenic risk may be generated. The likelihood of side effects increases in line with increases in the HQ value.

The HQ-prediction equation is as follows:HQ = (EPC·TWF)/RFC(4)
where EPC, TWF, and RFC denote exposure concentration, asbestos fibre concentration in the air under specific activity conditions (f·cc^−1^), and time weighting factor, respectively, referring to cumulative discontinuous exposure and reference intake concentration (f·cc^−1^) in a year.

## 3. Results and Discussion

### 3.1. Feature Analysis of Asbestos Contents in Environment Media from Different Areas

As depicted in Table 1, a total of 94 soil samples were collected. Asbestos was not detected in eight of the ninety-four soil samples, and asbestos was detected in all the rest samples, wherein the concentration of asbestos was in the range of 0.3–91.92% with an average value of 21.62%. In total, 57 ambient air samples were collected, and the detection results showed that 23 samples were below the detection limit. The airborne asbestos-fibre concentration was in the range of 0.008—0.145 f·cc^−1^, with an average value of 0.0224 f·cc^−1^. The large dispersion degree of the asbestos concentration in the sample was mainly correlated with the industrial production activities, the heterogeneity of the pollution media, and the migration of the pollutants.

As depicted in Figure 2a, the average contents of asbestos in soil samples from the mining area, the ore dressing area, the periphery of the waste pile, the residence area, and the bare land reached 39.06%, 36.84%, 19.92%, 6.69%, and 1.81%, respectively. The sampling results suggested that different pollution levels were present in the respective area, where the concentrations of asbestos in the soil in the dressing and mining areas were increased; the pollution levels at the periphery of the waste pile ranked second, and the pollution levels in the residential area and in the bare land were relatively light. The soil-quality standards in some countries stipulate the content of asbestos dust and establish a series of cleaning measures. Given the current level of testing technology, the USEPA recommends that cleaning measures should be employed when the content of asbestos in the soil is 0.25–1%. The UK has unofficially recommended that the critical value of asbestos pollutant content in the soil is 0.01%. Furthermore, the Australian Polluted Land Advisory Association recommends that the critical value of the asbestos content in the soil is 0.01%. Moreover, Finland has recommended that control measures should be taken when the content of chrysotile asbestos in the soil is 100 mg/kg (0.01%), or when the hornblende-asbestos count is 10 mg/kg (0.001%), and the intervention value for asbestos content in the soil developed by the Dutch stratified assessment method reaches 100 mg/kg (0.01%) [7,8,12,13,14,15]. Compared with the soil quality standards formulated by a wide range of countries, the concentrations of asbestos in the soil detected in this study all exceeded the standard, such that cleaning measures should be taken.

As depicted in Figure 2b, the average contents of asbestos in the air samples collected from the mining area, the ore-dressing area, the periphery of the waste pile, the residential area, and the bare land were 0.025, 0.050, 0.039, 0.023 and 0.012 f·cc^−1^, respectively. As indicated by the sampling results, the airborne asbestos in the ore-dressing area and on the periphery of the waste pile exhibited relatively high concentrations, the pollution levels in the mining area ranked second, and the pollution levels in the bare-land area were the lowest. Some foreign institutions have set limits for the air concentration of asbestos. For instance, the National Institute of Public Health and the Environment of the Netherlands proposed a hypothesis, in 1989, that the inducement of lung cancer is closely correlated with chrysotile asbestos, and their established air quality standard reaches 0.01 f·cc^−1^; in 1991, the Air Quality Standard set by the American Heat Exchanger Association was 5 × 10^−5^ f·cc^−1^. In 2000, the World Health Organization (WHO) formulated an air-quality standard of 10^−4^–10^−3^ f·cc^−1^, suggesting that the lifetime exposure risk of 5 × 10^−4^ f·cc^−1^ is 10^−6^–10^−5^. In 2010, the Dutch Health Council set an air quality standard for chrysotile asbestos in accordance with the results of the analysis of epidemiological data, reaching 2.8 × 10^−4^ f·cc^−1^. In 2014, the USEPA formulated an air-quality standard of 1.2 × 10^−4^ f·cc^−1^ [12] based on a toxicity-parameter inhalation-unit risk of 0.17 f·cc^−1^ and a reference intake of 9 × 10^−5^ f·cc^−1^. The occupational health standard established in China is 0.8 f·cc^−1^ [16]. The results obtained in this study exceeded the air-quality standards set by foreign institutions, and they were lower than the domestic occupational health standards.

Since open-pit mining areas are characterised by multiple dust-producing sites, large dust-production volumes, and high dust concentration in the air [17], the concentrations of asbestos in the soil and air are significantly correlated with the frequent occurrence of asbestos-related production activities (e.g., the excavation of mining pits, dressing workshops for production, raw-ore-crushing workshops, the dumping of waste-slag piles, transportation routes for the hauling of asbestos ore, and finished product vehicles). The average concentration of soil asbestos in the mining area was relatively high, reaching 39.06%. Unlike the pollution degree of soil asbestos, the area with the highest air concentration was the ore-dressing area, with a concentration of up to 0.050 f·cc^−1^, suggesting that the areas with frequent activities (e.g., asbestos-mine destruction, processing, and disposal activities) exhibited a prominent disturbance-and-release effect on the asbestos fibres.

### 3.2. Feature Analysis of Asbesto-Fibre Microtopographies in Different Area Environment Media

Figure 3 presents the SEM characterization results of the asbestos samples.The asbestos fibres in the raw ore were different in size, and generally present in the form of short columns and long strips. Some fine particles were randomly distributed on the surfaces of the fibres. The fibres were interlaced and closely bonded with each other. In the presence of the aforementioned interlaced structure, a mechanical binding force or a twisting force were generated between the asbestos fibres [18]. Asbestos was generally present in the tailings in the form of particles and elongated fibres, such that some irregularly shaped block particles and fewer fibres that were aggregated to form bundles or densely bound were observed. In the finished product, asbestos was essentially formed as flocculent and strip fibres, with sharp fracture surfaces and more fibres pulled out. Compared with the morphology distribution of the tailings, the amount of particles adsorbed onto the surface of the asbestos declined significantly, and the diameters of the particles also reduced. The asbestos was formed in the dust as particles with different sizes, and it was amorphous; dense particles completely covered the surfaces of the fibres, which agglomerated together. Furthermore, the gaps between the short columnar fibres were covered by particle agglomerates, with a relatively high particle coverage rate on the surfaces of the fibres.

The scanning morphologies of the different asbestos types suggested that different types of asbestos fibre exhibited typical structural characteristics. Most of the asbestos fibres in the raw ore were in the form of long strips, with a certain binding force but weak grain-boundary strength [19], and the fibres may have been easy to break after being pulled out due to the damage caused by the external force. Asbestos was obtained from the tailings, finished products, and dust after external stress was applied to it in the raw ore, and it generally took the form of strips, short columns and particles. Most of the fibres therein were no more than 5 μm in length and less than 3 μm in diameter; they were easy to diffuse and difficult decompose after entering the human lung [20].

Loomis et al. [21] suggested that the morphological characteristics of asbestos are correlated with its pulmonary pathogenicity. In general, irregular, thin, and long asbestos fibres can cause pulmonary fibrosis and increase the onset risk of lung cancer. Asbestos fibres with diameters of 0.5–5 μm and length-to-diameter ratios of over 3 can be more easily to enter the alveoli, thus posing a health hazard, while asbestos fibres with diameters of less than 0.5 μm are inhaled and immediately exhaled, and asbestos fibres with diameters of more than 5 μm are obstructed by nasal hairs [22]. By observing the morphologies of asbestos fibres under a scanning-electron microscope, the arrangement sequence of the fibres’ slenderness (the length-to-diameter ratio was higher than 3) in different samples was as follows, in descending order: asbestos finished products > asbestos raw ore > tailings > dust-removal dust. However, the asbestos contained in the raw ore was a bundle of more than 3000 fibres, which can be harmless before being worn and weathered into a single fibre, which is impossible to inhale into the human body through the respiratory tract [23]. Accordingly, asbestos dust close to the ore dressing area and the waste pile poses a relatively high risk to human health.

Furthermore, in this study, typical soil samples from different areas for SEM-imaging analysis were selected. As depicted in Figure 4, the asbestos fibres in the soil were essentially present in strip shapes, and the quantities of the fibres and aggregates showed a significant increasing trend with the increase in the asbestos in the soil. When the concentration of asbestos was low (0.46%), a small amount of asbestos fibre was attached to the surfaces of massive and granular soils. As the concentration of asbestos increased (up to 46.44%), irregular strip-shaped fibres agglomerated and displayed clearer profiles.

The soil asbestos in different areas had different morphologies and concentrations. The concentration of asbestos in the bare land was relatively low, and a small amount of asbestos fibre adhere to the surfaces of massive and granular soils; in the soil samples near a dressing plant, a large number of irregular strip-shaped fibres agglomerated, and some were flocculent.

### 3.3. Human-Health Risk of Airborne Asbestos Fibres

Given the effect of asbestos on human health in office areas, living areas, and surrounding residential areas in the mining area, 32 air-sampling sites were arranged in the ancillary areas exhibiting human activities (e.g., residential and office areas). Adults were the main sensitive receptors in the office and living areas in the mining area, whereas children and adults were the sensitive receptors in the peripheral residential areas. In general, asbestos in soil invades into the nearby atmosphere or the outdoor environments of residential and office areas through air media, such that inhalation was confirmed as a main exposure route of the human body.

Relevant calculation parameters were determined in exposure scenarios (Table 2), and the excess lifetime cancer risk and non-carcinogenic risk of inhaled airborne asbestos fibres were assessed using the Superfund asbestos risk-assessment-method. In this study, the parameters primarily originated from the Comprehensive Risk Information System of the USEPA, the International Agency for Cancer Research (IARC), the Superfund Asbestos Pollution Survey Framework, and Appendix E. The outdoor exposure frequency was set in accordance with the recommended parameters in the Technical Guidelines for Soil Pollution Risk Assessment on Construction Land (HJ25.3—2019) [7,24,25,26].

According to the on-site survey and personnel interviews, the living groups on the outskirts of the mining area and in the mining area were primarily migrant workers, the number of children in the study area was very low, with the majority of the population being adults, and these groups did not typically reside and work in the area for an extended period of time. Statistics indicate that the maximum lifespan was 22 years. In accordance with the extant IUR parameters of the the U.S. Superfund Risk Assessment Framework [7], the relatively recent 24 years were used as the calculation parameters for the conservative evaluation. Due to the poor natural conditions in this area and the surrounding desert terrain, it is impossible to engage in farming and outdoor recreation; therefore, it can be assumed that running and walking are the only outdoor activities conceivable, and the initial working age is 20 years. The 24 years of work and residence, constitute the exposure time for adults to various situations involving asbestos. In addition, there are no schools in the studied area, so children between the ages of 5 and 6 must travel elsewhere to attend school. This study therefore implies that the initial exposure age of children is 1 year old and that the duration of exposure is 5 years. Since data from the the U.S. Superfund Risk Assessment Framework indicate that the IUR parameter for children playing in soil is 0.045 [7], and this scenario represents the worst case of direct asbestos-contamination exposure, the above parameters represent the most conservative assessment.

A total of 32 sampling sites in the study area were subjected to the carcinogenic risk assessment, and the assessment results are presented in Figure 5. As indicated by the results, the ELCR of human health was at an acceptable level. As indicated by the parameter sensitivity analysis [26], if the 10% of daily exposure time changed, the parameter sensitivity (SR) was −91%, suggesting that the calculation results for the human-health carcinogenic risk can be closely correlated with the daily exposure time. The relatively low outdoor exposure of the population in this area resulted in a low overall assessed risk level.

Figure 5 presents the non-carcinogenic hazard indexes of the asbestos pollutants at air the sampling sites in the study area. As indicated by the results, the non-carcinogenic hazard indexes in 13 of the 32 sites exceeded 1, and there were sites exceeding the human health risk level in each region. In most cases, the sites with a high hazard index (HQ) were located in the vicinity of the tailing slag piles and dressing plants in the downwind direction, and they were notably affected by monsoon dust, with a maximum hazard index of 5.94. In general, except for the mining area, the rank of the average non-carcinogenic risk level in the respective area was as follows, in descending order: the periphery of the waste pile > ore-dressing area > residential area > bare land.

The high-risk sites in this study were located around the waste pile and the ore dressing area, and they were closely correlated with frequent production activities and greater degree of asbestos-fibre escape, and the corresponding average non-carcinogenic-risk level was high, further verifying the effect of the peripheral environment and dust morphology on human health. The residential area and the bare land were far from the scope of the production activities, and no waste piles or asbestos products were close to the residential area or the bare land. Furthermore, most of the sites were located in the dominant upwind direction, with a relatively low average non-carcinogenic-risk level. Yang Wenfang et al. [29] performed a remote-sensing extraction test of asbestos-dust pollution in the Ruoqiang-asbestos mining area. The results suggested that asbestos-dust pollution was distributed throughout the entire activity area and centred on the asbestos-mining and ore-dressing areas, which was consistent with the results of this study.

According to the constructed risk-assessment model, the asbestos-pollutant-risk-control value based on the protection of human health was obtained using a non-carcinogenic hazard index of 1 and an excess lifetime cancer risk of 10^−4^ as acceptable risks. As indicated by the results, the carcinogenic-risk-control values in the air were 0.1438, 0.2225 and 0.1540 f·cc^−1^ for adults working or living in the mining areas, adults’ outdoor activities in peripheral residence areas, and children’s outdoor activities, respectively; the non-carcinogenic-risk-control values in the air in the three scenarios reached 0.0084, 0.0090, and 0.0090 f·cc^−1^, respectively.

The human-health-risk-assessment method was adopted earlier, in accordance with the European and American air-quality standards. The European and American authorities believed that the inducement of lung cancer and pleural thickening was closely correlated with chrysotile asbestos, based on IRIS toxicological parameters, epidemiological data and the maximum carcinogenic or non-carcinogenic risk level. Given the differences between the parameters and maximum risk levels employed by different institutions, the derived quality standards differed significantly. For instance, the USEPA recommended unit inhalation risks of 0.23 f·cc^−1^ and 0.17 f·cc^−1^ in 1988 and 2014, respectively. The Netherlands National Institute of Public Health and the Environment considers that an unacceptable risk level is 10^−4^, and the USEPA set it at 10^−5^. As depicted in Table 3, the range of air quality guidelines for each institution was 5 × 10^−5^–0.01 f·cc^−1^, and the non-carcinogenic risk control value calculated in this study was only lower than the air-quality standard of 0.01 f·cc^−1^ established by the National Institute of Public Health and Environment of the Netherlands; it was higher than the values established by other institutions.

China has formulated a relatively loose sanitary standard for asbestos in the production environment (0.8 f·cc^−1^), whereas the allowable standard for asbestos dust in the living environment has not been specified thus far [16]. Thus, the risk control value calculated in this study can be used as a guide value for air quality under office or residential conditions in a mining environment.

At present, given the limitations of asbestos-land management, the complexity of exposure scenarios, and the universality of assessment methods, China has not formulated soil-asbestos-standard values thus far, which will be considered when conditions are mature. A database of considerable measurement data can be built through “worst-case” box simulation experiments and daily practices. The Netherlands [14] used soil and air-content data for linear simulation analyses. Moreover, a soil-asbestos screening value was developed through deduction based on the calculated risk-control value of airborne asbestos. Lastly, the intervention value was determined as 100 mg/kg. China’s relevant specifications for the health-risk assessment of polluted lands are undergoing progressive modification. The deduction method of the soil-asbestos limit values in the Netherlands is of critical significance in supporting the formulation of soil-asbestos-screening values in China.

## 4. Conclusions

(1)The soil surface and air in the sampling area has been polluted to different extents. The asbestos concentration on the soil surface in the mining area was the highest, and those in the ore-dressing area and the waste-pile area were ranked second. The ore-dressing area exhibited the highest asbestos air concentration.(2)The SEM-imaging results showed that the main morphologies of the asbestos were basically strip, short columnar, and granular. The arrangement sequence of the slenderness (length-to-diameter ratio) of the asbestos fibre was as follows, in descending order: asbestos products > asbestos ore > tailings > dust.(3)The release of asbestos dust close to the dressing and tailing areas directly led to the increase in the concentration of asbestos in the polluted soil. The irregular strip-shaped fibres were agglomerated, such that the profile turned out to be clearer.(4)The risk-control values in the three scenarios (adults working in offices or living in the mining area, adults’ outdoor activities in the peripheral residentia area, and children’s outdoor activities) reached 0.0084, 0.0090, and 0.0090 f·cc^−1^, respectively. In total, 40.6% of the sites exceeded the unacceptable non-carcinogenic risk level. Most of the sites exhibiting higher risk indexes were located close to the waste pile in the downwind direction.

## Figures and Tables

**Figure 1 toxics-11-00494-f001:**
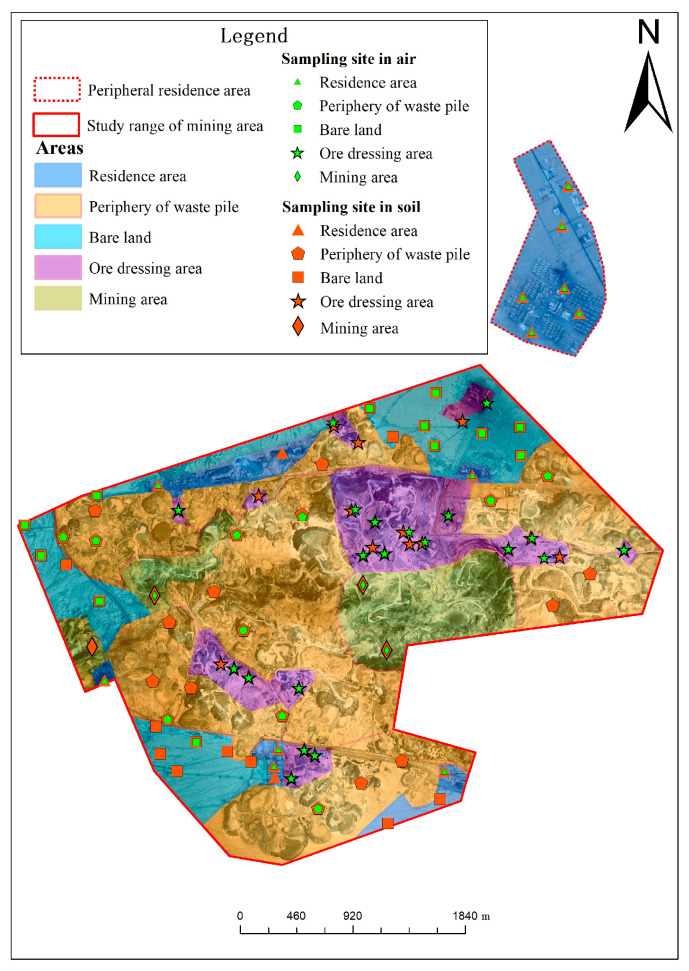
Distribution map of air and soil sampling sites.

**Figure 2 toxics-11-00494-f002:**
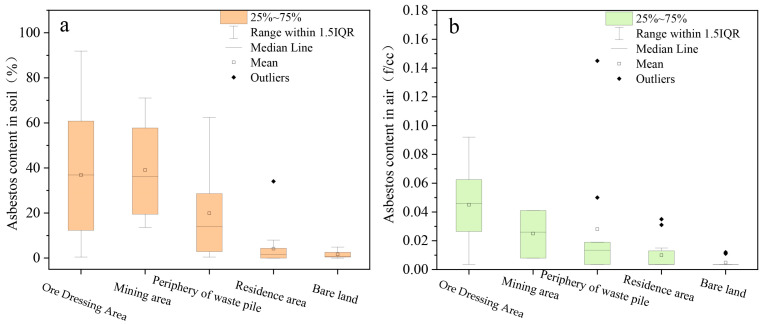
Statistical results of asbestos content in soil and air samples. (**a**): asbestos content in soil; (**b**): asbestos content in air.

**Figure 3 toxics-11-00494-f003:**
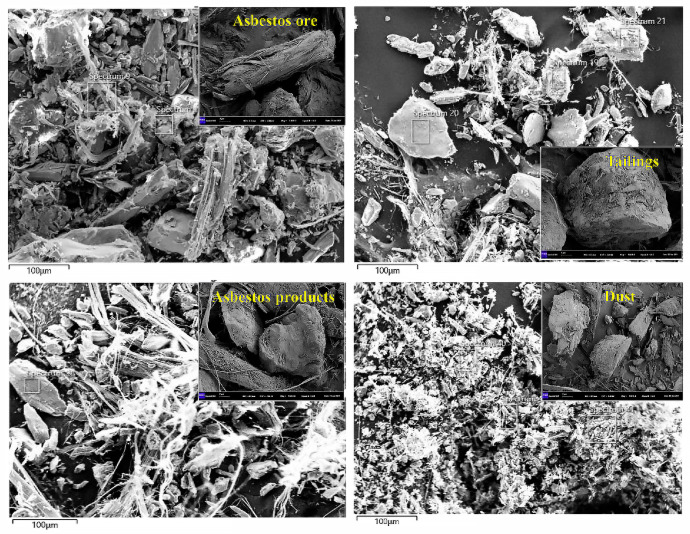
SEM morphology-characterization results of different asbestos samples.

**Figure 4 toxics-11-00494-f004:**
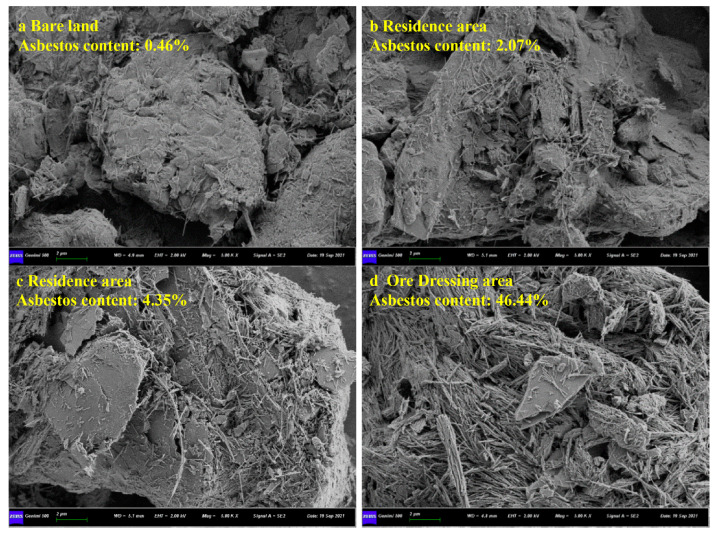
SEM imaging results of asbestos in soil.

**Figure 5 toxics-11-00494-f005:**
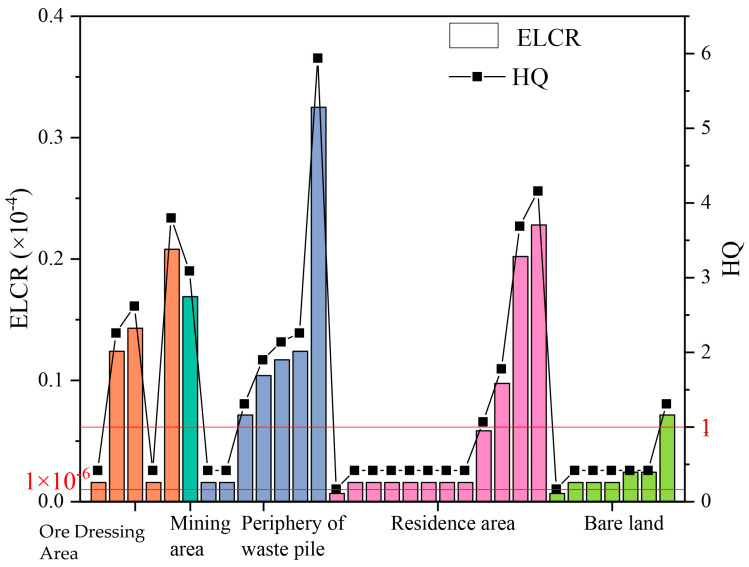
The results of human-health-risk assessment of airborne asbestos.

**Table 1 toxics-11-00494-t001:** Descriptive Statistics of Asbestos in Surface Soil and Air.

Type of Environment Media	Sampling Area	Number of Tests (Piece)	Number of Detections (Piece)	Median	Minimum	Maximum	Average	Standard Deviation
Soil asbestos (%)	Mining area	6	6	36.19	13.65	71.06	39.06	21.29
	The periphery of waste pile	21	21	14.01	0.46	62.43	19.92	18.95
	Residence area	16	10	3.53	1.69	34.04	6.69	8.06
	Bare land	21	19	1.87	0.30	4.87	1.81	1.45
	Ore-dressing area	30	30	36.90	0.42	91.92	36.84	26.83
Airborne asbestos (fibre/cm^3^ & f·cc^−1^)	Mining area	3	3	0.026	0.008	0.041	0.025	0.013
	The periphery of waste pile	10	7	0.018	0.011	0.145	0.039	0.042
	Residence area	12	4	0.023	0.011	0.035	0.023	0.012
	Bare land	12	2	0.012	0.011	0.012	0.012	0.004
	Ore-dressing area	20	18	0.050	0.019	0.092	0.050	0.024

**Table 2 toxics-11-00494-t002:** Parameters and values of human health risk assessment model for airborne asbestos.

Parameters of Assessment Model	Office and Living Areas in the Mining Area	Peripheral Residential Areas		Value Source
Sensitive receptor	Adult	Children	Adult	
Outdoor-exposure frequency*/(d·a^−1^)	62.5	87.5	87.5	[5]
Daily exposure time/h	1.5	1	1	Actual survey
Time-weighting factor (TWF)	0.0107	0.01	0.01	[3,27,28]
Inhalation-unit risk (IUR)	0.065	0.045	0.065	[3,27,28]
Reference concentration (RFC)/(f·cc^−1^)	0.00009	0.00009	0.00009	[11]
Note	Age of initial exposure: 20 years old, exposure period: 24 years	Age of initial exposure: 1 year old, exposure period: 5 years	Age of initial exposure: 20 years old, exposure period: 24 years	[3,27,28]

Note: *outdoor exposure frequency represents a recommended land use value in the Technical Guide for Soil Environmental Investigation and Assessment of Construction Land (HJ25.3—2019), the adult exposure frequency in the mining area employs the second class of land value, and the peripheral residence area applies the first class of land value.

**Table 3 toxics-11-00494-t003:** Published air-quality standards.

Agency/Time	Air Quality Guideline (f·cc^−1^)	Health Risk Level/Corresponding Concentration (f·cc^−1^)	Standard Note
USEPA (1988) [30]	8 × 10^−5^	10^−5^/40	IUR is 0.23 f·cc^−1^
Netherlands National Institute for Public Health and the Environment (1989) [31]	1 × 10^−2^	10^−4^	The maximum carcinogenic risk level is 10^−4^, values of less than 10^−6^ indicate no risk.
American Heat Exchanger Association (1991) [32]	5 × 10^−5^	4 × 10^−5^/100	ELCR is 4 × 10^−5^, equivalent to a concentration of airborne asbestos of 100 f·cc^−1^
WHO (2000) [33]	1–10 × 10^−4^	10^−6^–10^−5^/5 × 10^−4^	5 × 10^−4^ f·cc^−1^ is equivalent to ELCR of 10^−6^–10^−5^
USEPA (2008) [7]	2 × 10^−4^	10^−5^/200	2 × 10^−4^ f·cc^−1^ is a guidance value calculated for a 30-year ELCR of 10^−5^
Dutch Health Council (2010) [34]	2.8 × 10^−4^	5 × 10^−6^/280	ELCR is 5 × 10^−6^, risk-control value is 280 f·cc^−1^
USEPA (2014) [35]	1.2 × 10^−4^	1/9 × 10^−5^	Inhalation-unit-risk value is 0.17 f·cc^−1^, critical reference concentration for non-carcinogenic risk is 9 × 10^−5^ f·cc^−1^

## Data Availability

Data will be made available on request.

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
