# Peer review of "Asbestos-Environment Pollution Characteristics and Health-Risk Assessment in Typical Asbestos-Mining Area"

_toxics, 2023, doi:10.3390/toxics11060494_

Round 1
Reviewer 1 Report
No major issue was found in the manuscript but English editing would be necessary to make it clear. In addition, more detailed information would be necessary for method section.
Introduction: it would be helpful to add regarding asbestos mining activities in China. It seems that asbestos mining is not banned in China.
Line 39 – 42 two sentences are deliberating the same information.
Line 43 I type carcinogens: please check whether it is a right term. IARC usually uses as “Group 1 carcinogens”.
Line 70 RBCA should be spelled out.
Line 126-132 Detail information about air sampling including sampler, flow rate, personal/area sampling, duration and standard method procedure are missing. For example, the filter that was used for the air sampling should be a mixed cellulose ester filter. It would be better to have detailed information regarding Chinese asbestos standard procedure because not many readers are familiarized with the standards; GBZ/T 192.5-2007 and GB/T 23263-2009.
Line 140 Was a phase contrast microscope used for counting? What was the counting rule? “the counting was observed under a microscope…” is not clear sentence. It might be “the fibers were counted with a microscope….”
Line 141-143 “whether the asbestos fibres observed under the microscope conformed to the counting requirements was more effectively verified using an ocular micrometer” is not clear.
Line 143 what was the equation?
Line 147 polarized light microscope should be the right term.
Line 151-157 is this method for air or soil samples? it was involved with crushing. What was the criteria between fibre and non-fibre particles? How did you differentiate between two.
Line 173-174 Is cumulative discontinuous exposure in a year time weighting factor? Confusing.
Line 196 23 sites not “samples”? It started with “57 ambient air samples”.
Line 204 – 207 it is not clear what authors wanted to say.
Line 257-283 it is not clear that sampling point of the SEM samples, not same as the soil and air samples and it was not described in the method section. why? In addition, only sample description was presented without any size distribution data.
Line 308 data is missing. Significant different can’t be seen in Figure 4.
Please see comments.
Reviewer 2 Report
This is an interesting study and the authors have made a good attempt at investigating the human risk of asbestos contamination in a mining area. However, I have a few comments/questions
1. The main issue I have is with how the USEPA framework was used, especially the choice of exposure duration (24 years for adults, both peripheral residential and living in mining area; 5 years for children). The choice of these need more explanation. In the USEPA document, 5-years for children is based on 5-years of playing in contaminated soil (@2hours per day, 350 days per week). Is this what the authors of this study assumed, or where they calculating a life-time risk, which is more likely to be continuous exposure over the period they are likely to live in the contaminated area. For adults they have used 24 years. However, in the framework 24 years is for recreational (running/walking) exposure. Baseline residential is 30 years in the USEPA framework example. I would suggest for this calculation the authors should estimate the average residential time in the area and apply that as the exposure period, or better explain why they chose the exposure periods they did.
2. Soil asbestos levels were reported as percentage. Is this weight/weight? The concentrations were extremely high. Indeed up to 92%. Weight/weight would mean that 92% of the sample is asbestos – is this correct? The authors suggest that pollution levels in the residence area and bare land were relatively light. I would suggest a median (not medium) of 3.53 and 1.69%, respectively, that these are heavily contaminated (just less so than the other areas).
3. What is meant be ‘dressing area’? I have never heard that term before.
4. What is meant by ‘continental plateau climate’ (line 107)? Do the authors mean ‘continental climate’
5. Sentence on lines 81/82 doesn’t make sense. Im not sure what ‘activity crowd’ means
6. Table 2, change ‘medium’ to ‘median’
7. Is the ‘risk control values’ the airborne fibre concentration that keeps the life-time risk below 10-4 on these calculations? I have heard the term ‘risk control value’ before.
8. Is air and soil sampling and analyses methods consistent with international methods? I assume so but there is not much detail.
9. Was there a minimum sampling time for air sampling?
10. The use of ‘acceptable’ risk is subjective and varies across international jurisdictions. Is this the acceptable risk in China?
There are a few grammatical errors but the quality of the English is mostly fine
Round 2
Reviewer 2 Report
The authors have addressed most of my comments. A few minor issues from the responses below
Point 1, the authors need to explain the assumptions (eg exposure duration, frequency, daily exposure time) in the text, or in a supplement. It is rather fortuitous that these values were the same for the residents/workers in this study as the example in the USEPA framework. I don’t question that the authors considered these issues for the relevant population but they need to justify using those values in the manuscript.
Point 5, the sentence needs to be reworded. I understand the authors response to me but the sentence still doesn’t make sense.
Point 8, are the Chinese standards and methods similar to other standards and methods (eg NIOSH). It would be good to know that they don’t vary from these.
The quality of English is mostly okay but the manuscript does need some editing to improve this. I havent identified all places that this could be improved but have included a couple of examples below
1. Sentence on lines 81/82 (point 5 in original review)
2. Remove ‘On that basis,’ – line 38
3. Remove ‘According to statistics,’ on line 40
4. Don’t refer to asbestos as ‘the asbestos’. Just refer to it as ‘asbestos’ (examples lines 43, 45, 48)
